# CO_2_ Adsorption on the N- and P-Modified Mesoporous Silicas

**DOI:** 10.3390/nano12071224

**Published:** 2022-04-05

**Authors:** Oyundari Tumurbaatar, Hristina Lazarova, Margarita Popova, Violeta Mitova, Pavletta Shestakova, Neli Koseva

**Affiliations:** 1Institute of Organic Chemistry with Centre of Phytochemistry, Bulgarian Academy of Sciences, Acad. G. Bonchev St., bl. 9, 1113 Sofia, Bulgaria; oyundari.tumurbaatar@orgchm.bas.bg (O.T.); hristina.lazarova@orgchm.bas.bg (H.L.); margarita.popova@orgchm.bas.bg (M.P.); pavletta.shestakova@orgchm.bas.bg (P.S.); 2Institute of Polymers, Bulgarian Academy of Sciences, Acad. G. Bonchev St., bl. 103A, 1113 Sofia, Bulgaria; mitova@polymer.bas.bg

**Keywords:** CO_2_ capture, modified mesoporous silicas, Schiff base, aminophosphonate

## Abstract

SBA-15 and MCM-48 mesoporous silicas were modified with functionalized (3-aminopropyl)triethoxysilane (APTES) by using the post-synthesis method, thus introducing N- and P-containing groups to the pore surface. The structure of the newly synthesized modifiers (aldimine and aminophosphonate derivatives of (3-aminopropyl)triethoxysilane and their grafting onto the porous matrix were proved by applying multinuclear NMR and FTIR spectroscopies. The content of the grafted functional groups was determined via thermogravimetric analysis. The physicochemical properties of the adsorbent samples were studied by nitrogen physisorption and UV–Vis spectroscopy. The adsorption capacity of CO_2_ was measured in a dynamic CO_2_ adsorption regime. The modified silicas displayed an enhanced adsorption capacity compared to the initial material. The ^13^C NMR spectra with high-power proton decoupling proved the presence of physically captured CO_2_. A value of 4.60 mmol/g was achieved for the MCM-48 material grafted with the Schiff base residues. The total CO_2_ desorption was achieved at 40 °C. A slight decrease of about 5% in CO_2_ adsorption capacities was registered for the modified silicas in three adsorption/desorption cycles, indicating their performance stability.

## 1. Introduction

Since the industrial revolution, the concentration of the greenhouse gases in the atmosphere has been significantly increased, and this affects the average temperature of our planet and global climate. For the elimination of greenhouse gases, much attention has been focused on improving the activity of absorbents for the capture of CO_2_. The adsorbent must have high selectivity and a high adsorption capacity for carbon dioxide at high temperatures, adequate adsorption/desorption kinetics for carbon dioxide at operating conditions, stable adsorption capacity of CO_2_ after repeated adsorption/desorption cycles and adequate mechanical strength of the adsorbent matrix after cyclic exposure to high-pressure streams [1]. Adsorbents can be classified into two groups: physisorbents and chemisorbents [2]. Physisorbents act as molecular sieves and adsorb CO_2_ onto their surfaces, while chemisorbents can chemically react with CO_2_. The zeolites [3], zeolitic imidazole frameworks [4], carbon-based materials [5], porous organic polymers [6] and amine functionalized materials [7,8,9] are among the adsorbents studied for CO_2_ capture. Ordered mesoporous silica, such as MCM- and SBA-types, are ideal solid adsorbents or supports of active adsorption sites with a high surface area, large pores and tunable pore sizes [8]. The mesoporous silica materials also possess a large number of silanol groups, which are a key precondition for suitable functionalization with different organic molecules, which improve silica’s performance in CO_2_ adsorption [9]. A variety of amine compounds, such as monoethanolamine (MEA), diethanolamine (DEA), tetraethylene pentamine (TEPA), aminopropyl trialkoxy siliane (APTS), PEI, 1-metylpiperazine, etc., have been identified and studied in sorbent modification. The most-often-applied techniques include impregnation or post-synthesis grafting of functional compounds [10,11,12,13,14].

Mesoporous MCM-41 impregnated with PEI demonstrates a higher capacity for CO_2_ than commercial sorbents, such as activated carbon and zeolites [15]. Nanoporous sorbent SBA-15 modified with 50% PEI achieved a sorption capacity of 3.18 mmol/g at 75 °C at a CO_2_ partial pressure of 0.15 bar [16]. In addition to increasing the adsorption capacity, amino-impregnation of mesoporous silicate has been applied to obtain sorbents that do not lose their capacity in the presence of moisture; on the contrary, moisture improves their capacity [17]. Due to its high resistance to moisture and thermal stability, porous alumina has also been studied as a potential sorbent for CO_2_ after modification with organic amines. Aluminum oxide impregnated with 40% DETA showed a capacity of 1.83 mmol/g at 25 °C [18]. Modified commercially available Y-type zeolite (Si/Al = 60) by impregnation with TEPA achieved an adsorption capacity of 4.27 mmol/g at 60 °C in the presence of 15% CO_2_ and 7% water vapor in the gas stream [19]. The zeolite impregnated with TEPA has been completely regenerated after 10 consecutive cycles of adsorption/desorption [20].

A grafting technique for the modification of mesoporous silica by interaction with amino silanes that results in covalently bound amine to the surface of the sorbent has also been used to obtain sorbents with a high capacity and improved water-vapor tolerance [14,21]. Interestingly, the combination of pore expansion and optimum grafting conditions increased the amine loading from 6.0 mmol/g for triamine grafted on regular MCM-41 in dry toluene under reflux (conventional conditions) to 8.0 mmol/g for triamine grafted on PE-MCM-41. The application of amino-functionalized SBA-type adsorbents also displayed promising results in the CO_2_ retention. An adsorption capacity of 0.72 mmol/g under 101.32 kPa at 60 °C was determined for SBA-16 modified with N-(2-aminoethyl)-3-aminopropyltrimethoxysilane. The adsorbent displayed a high hydrothermal stability, and the combustion of the amino-groups in air was observed at temperatures above 200 °C [22]. Hiyoshi et al. [23] reported that the adsorption capacity of amino-silica (SBA-15) reached 1.8 mmol/g under 15 kPa CO_2_ at 60 °C and found that the efficiency of adsorption increased with increasing the surface density of amine [24].

Similarly, triamine-functionalized SBA-15 had a greater CO_2_ retention capacity value than the grafted with mono- and di-amines matrices. The presence of moisture enhanced the adsorbent performance, suggesting the participation of water molecules in CO_2_ retention [25]. Moreover, 3-aminopropyltri(m)ethoxisilanes (APMES or APTES) were used for the functionalization of plant-derived sorbents [26,27]. Mesoporous SBA-15 synthesized from rice husk ash was grafted with APTMS for the subsequent growth of tris(2-aminoethyl) amine (TREN) dendrimers. A CO_2_ adsorption capacity of 5–6 and 7–8 wt.% was observed for the material with second and third dendrimer generation, which was higher than the reported values for melamine and PAMAM dendrimers [26]. APTES was used for the modification of cocoa-shell-based hydrochar (HC), followed by cobalt particle incorporation. The obtained HC-APTES-Co sorbent displayed a higher CO_2_ retention capacity than its metal-free counterpart, in spite of its lower basicity and porosity. The material afforded a high surface affinity toward CO_2_, though the direct contact between CO_2_ and the amino groups was hampered. The achieved result was the reversible capture of CO_2_ with easy regeneration due to predominant physical CO_2_ condensation [27].

The molecular simulation studies have been applied to evaluate the effect of different factors on the adsorbent CO_2_ capture performance. The computationally explored parameters include surface defects, temperature, humidity, adsorbate composition, etc. The results of the simulation studies provide a comprehensive understanding of the surface interactions and assistance in the design of the adsorbent material [28,29].

The research efforts focused on the development of adsorbents with an enhanced affinity to CO_2_ molecules led to extending the range of techniques and functionalities used for surface modification. Besides amino groups, other functionalities, such as acetyl, formyl and nitro groups, were introduced in a series of triptycene-based three-dimensional rigid framework that displayed a good CO_2_ adsorption capacity and a high CO_2_/N_2_ selectivity [30].

In the present study, new derivatives of (3-aminopropyl) triethoxysilane (APTES) were synthesized and used in the post-synthetic modification of ordered mesoporous silica. The first compound presented a Schiff base derived from APTES and furfural. The second agent contained an aminophosphonic segment. The modified SBA-15 and MCM-48 were studied in CO_2_ capture experiments to evaluate their features as adsorbents.

## 2. Materials and Methods

The experimental design is presented in Figure 1. It included the synthetic steps to obtaining the modifying agents derived from APTES and the preparation of the adsorbents followed by their functionalization. At each synthetic step, the product was characterized by applying appropriate physicochemical methods.

### 2.1. Materials

Pluronic P123 (≥98%) (Sigma-Aldrich Chemie, Schnelldorf, Germany), (3-aminopropyl)triethoxysilane (APTES) (≥98%) (Sigma-Aldrich, Saint Louis, MO, USA), hexadecyltrimethylammonium bromide (CTAB) (≥98%) (Sigma-Aldrich Chemie, Schnelldorf, Germany), tetraethyl orthosilicate (TEOS) 98% (Fisher Scientific, Loughborough, UK), hydrochloric acid (37%) (Valerus, Bulgaria) and sodium hydroxide (NaOH) (Merck, Darmstadt, Germany) were used without further purification. Furfural (≥98%) and diethyl H-phosphonate (≥95%) were Sigma-Aldrich (Sigma-Aldrich Production GmbH, Product Brand Fluka, Buchs, Switzerland) products and were distilled prior to use. Toluene and diethylene glycol dimethyl ether (diglyme) (Sigma-Aldrich, Saint Louis, MO, USA) were dried via azeotropic distillation. Deionized distilled water was used in the preparation of all solutions.

### 2.2. Preparation of SBA-15 and MCM-48

SBA-15 and MCM-48 were prepared by hydrothermal synthesis. The synthesis procedure for MCM-48 consisted of the following steps: (a) dissolving of cetrimonium bromide (8.8 g, 24.1 mmol) (CTAB) in water (80 mL) at 35 °C and, after that, adding 10 mL of 2 M NaOH and 10 mL (9.33 g; 44.8 mmol) of tetraethyl orthosilicate and stirring the mixture at room temperature for 1.5 h; and (b) transferring the homogeneous solution to a Teflon vessel jacketed in a stainless-steel autoclave and heating in an oven at 80 °C for 72 h. The product was filtered and then calcined in an oven at 300 °C for 2 h and at 550 °C for 8 h.

The synthesis of SBA-15 was performed by using the following the procedure. A solution of Pluronic P123 (6 g, 1.0 mmol) in 1.5 M HCl (120 mL) was added at room temperature to a solution of CTAB (1.2 g, 3.3 mmol) in 50 mL H_2_O. The mixture was then dropped by means of a dropping funnel to a solution of 20 mL (18.66 g; 89.6 mmol) tetraethyl orthosilicate (TEOS) in ethanol (40 mL) and stirred at 35 °C for 45 min. The homogeneous solution was transferred to a Teflon vessel jacketed in a stainless-steel autoclave and heated in an oven at 75 °C for 16 h and at 120 °C for 36 h. The product was filtered and was calcined in an oven at 300 °C for 2 h and at 550 °C for 5 h.

### 2.3. Preparation of a Schiff Base from APTES and Furfural

Furfural (0.86 g, 9.0 mmol) and APTES (2.00 g, 9.0 mmol) were dissolved in toluene (10 mL) and stirred for 24 h at room temperature. The obtained Schiff base was isolated under reduced pressure. The product was assigned as SAPTES.

^1^H NMR (600 MHz, CDCl_3_) *δ* ppm: 8.00 (s, 1H, –N=CH–); 7.428 (d, 1H, FurH-5); 6.644 (dd, 1H, FurH-3); 6.394 (dd, 1H, FurH-4); 3.746 (q, 6H, CH_3_CH_2_O–); 3.52–3.49 (m, 2H, –CH_2_CH_2_N–); 1.80–1.74 (m, 2H, –CH_2_CH_2_CH_2_Si–); 1.147 (t, 9H, CH_3_CH_2_O–);0.597–0.569 (m, 2H, –CH_2_CH_2_Si–).

^13^C{H} NMR (600 MHz, CDCl_3_) *δ* (ppm): 151.58 (–CH=N); 149.61 (FurC-2); 144.52 (FurC-5); 113.63 (FurC-3); 111.49 (FurC-4); 64.48 (=NCH_2_–); 58.34 (–OCH_2_CH_3_); 24.17 (=NCH_2_CH_2_CH_2_–); 18.26 (–OCH_2_CH_3_); 8.01 (–SiCH_2_–).

IR (cm^−1^): 2974–2829 (ν(C–H)); 1643 (ν(N=CH)); 1100–1014 (ν(C–O)) and ν(Si–OCH_2_)); 750 γ(FurC–H).

### 2.4. Preparation of Aminophosphonate Derivative of APTES

SAPTES (1.82 g, 6.1 mmol) and diethyl H-phosphonate (0.88 g, 6.4 mmol) were dissolved in diglyme (5 mL). The catalyst CdI_2_ (0.046 g, 0.13 mmol) was added, and the reaction mixture was stirred for 24 h at 50 °C. The obtained product was assigned as PAPTES.

^1^H NMR (600 MHz, CDCl_3_) *δ* ppm: 7.31 (s, 1H, FurH-5); 6.27 (s, 2H, FurH-4 and FurH-3); 4.10–3.95 (m, 4H, CH_3_CH_2_OP(O)–); 3.87–3.83 (m, 1H, –HNCH(Fur)P(O)–); 3.695 and 3.62 (q, 6H, CH_3_CH_2_OSi–); 2.53–2.40 (m, 2H, –CH_2_NH–); 1.24–1.10 (m, 17H, –CH_2_–CH_2_–CH_2_–Si– and CH_3_CH_2_–O–P(O)– and CH_3_CH_2_OSi–); 0.53–0.52 (m, 2H, –CH_2_–CH_2_–CH_2_–Si–).

^13^C NMR (600 MHz, CDCl_3_) *δ* ppm: 150.13 (s, 1H, FurC-2); 142.36 (d, ^4^J=2.7 Hz, 1H, FurC-5); 110.54 (d, ^4^J = 2.7 Hz, 1H, FurH-4); 108.90 (d, ^3^J = 7.3 Hz, 1H, FurC-3); 63.16 and 62.85 (2d, ^2^J = 6.7Hz, CH_3_CH_2_OP(O)–); 58.33 (s, CH_3_CH_2_OSi–); 54.61 (d, ^1^J = 160.4 Hz, –HNCH(Fur)P(O)–); 50.98 (d, ^3^J = 15.3 Hz, –CH_2_NHCH(Fur)P(O)–); 23.02 (s, –CH_2_CH_2_NHCH(Fur)P(O)–); 18.29 (s, CH_3_CH_2_OSi–); 16.47 and 16.34 (2d, ^3^J = 5.5 Hz, CH_3_CH_2_OP(O)–); 7.77 (s, –CH_2_–CH_2_–CH_2_–Si–).

^31^P{H} NMR (600 MHz, CDCl_3_) δ ppm: 21.12.

^31^P NMR (600 MHz, CDCl_3_) δ ppm: 21.12 (sextet).

IR (cm^−1^): 2976–2818 (ν(C–H)); 1250 (ν(P=O)); 1103–1024 (ν(C–O)); ν(P–OCH_2_) and ν(Si–OCH_2_)); 750 γ(FurC–H).

### 2.5. Modification of MCM-48 and SBA-15 with SAPTES

SBA-15 or MCM-48 (0.5 g) was heated in an oven at 120 °C for 1 h. The hot SBA-15 or MCM-48 was dispersed in toluene (50 mL). After that, SAPTES (0.97 g; 3.2 mmol) was added to the mixture. The mixture was stirred for 24 h at 60 °C on a magnetic stirrer; after that, the product was washed three times with chloroform to remove the unreacted modifying agent. The modified samples were denoted as SBA-15/SAPTES and MCM-48/SAPTES.

### 2.6. Modification of SBA-15 and MCM-48 with PAPTES

SBA-15 or MCM-48 (0.5 g) was heated in an oven at 120 ° C for 1 h. The hot SBA-15 or MCM-48 was dispersed in toluene (50 mL). After that, PAPTES (1.42 g; 3.2 mmol) was added to the mixture at 60 °C. The mixture was stirred for 24 h at 60 °C on a magnetic stirrer; after that, the product was washed three times with chloroform to remove any unreacted modifying agent. The modified samples were denoted as SBA-15/PAPTES and MCM-48/PAPTES.

### 2.7. Methods

#### 2.7.1. Materials Characterization

NMR spectra were recorded on a Bruker Avance II+ 600 NMR spectrometer (Bruker, Germany) operating at 600.01 MHz ^1^H frequency (119.21 MHz for ^29^Si, 241.88 MHz for ^31^P), using 4 mm dual ^1^H/(^15^N-^31^P) solid-state CP/MAS (CP = Cross-Polarization, MAS = Magic Angle Spinning) probehead (Bruker, Germany). The samples were loaded in 4 mm zirconia rotors and spun at an MAS rate of 10 kHz for all measurements. The ^1^H → ^29^Si and ^1^H → ^13^C CP MAS spectra were measured with the following experimental parameters: ^1^H 90° pulse length of 3.6 μs, 2 ms contact time, 5 s relaxation delay, typically 6000 scans for ^1^H → ^29^Si and 2000 scans for → ^1^H → ^13^C spectra. The ^1^H SPINAL-64 decoupling scheme was used during the acquisition of CP experiments. ^13^C HPDEC NMR spectra were measured with a 90° pulse length of 4.6 μs, a recycle delay of 60 s, a power level of 80 kHz for 1H decoupling during acquisition was used and typically 512–1024 scans were accumulated. The ^31^P spectra were measured with a direct excitation (single pulse experiment) with the following experimental parameters: ^31^P 90° pulse length of 8.5 μs, relaxation delay of 300 s and 128 scans. An exponential window function was applied before Fourier transformation (line broadening factor 50), and the data were zero-filled to 16 K data points for all spectra.

ATR–FTIR spectra were recorded by means of an IRAffinity-1 “Shimadzu” Fourier-Transform Infrared (FTIR) spectrophotometer (Shimadzu, Japan)with MIRacle Attenuated Total Reflectance Attachment. The instrument was equipped with temperature-controlled, high-sensitivity DLATGS detector and ATR attachment with KRS-5 prism. In general, 50 scans and a 4 cm^−1^ resolution were applied. The spectral data were processed with IRsolution software.

The thermogravimetric measurements were performed with a STA449F5 Jupiter of NETZSCH Gerätebau GmbH (Netzsch, Germany) up to 600 °C, with a heating rate of 5 °C/min in air flow, followed by a hold-up of 1 h.

Determination of the specific surface by Brunauer, Emmett and Teller (BET); the diameter; and the pore size distribution of the obtained materials were performed by low-temperature nitrogen adsorption. The adsorption and desorption isotherms of nitrogen at −196 °C were determined in the pressure range p/p_0_ = 0.001–1, using an advanced micropore size and chemisorption analyzer “AUTOSORB iQ-MP/AG” (Anton Paar GmbH, Austria), with a heating rate of 5 °C/min in air flow. Before every measurement, the samples were degassed at 80 °C for 16 h.

#### 2.7.2. CO_2_ Adsorption

CO_2_ adsorption experiments were performed in dynamic conditions in a flow system. The sample (0.40 g adsorbent) was dried at 150 °C for 1 h, and 3 vol.% CO_2_/N_2_ at a flow rate of 30 mL/min was applied for the adsorption experiments at 25 °C. The gas was analyzed online by using gas chromatograph NEXIS GC-2030 ATF (Shimadzu, Japan) with 25 m PLOT Q capillary column.

CO_2_ adsorption measurements under static conditions were measured at 0 °C with an AUTOSORB iQ-MP-AG (Anton Paar GmbH, Graz, Austria) surface area and pore size analyzer (from Quantachrome, Anton Paar GmbH, Graz, Austria). The sample powders were weighed on an analytical balance. Transformation of the primary adsorption data was performed by the Quantachrome software.

## 3. Results and Discussion

The present investigation was aimed at enhancing the carbon dioxide adsorption capacity of mesoporous silicas (SBA-15 and MCM-48) via surface modification with phosphorus- and nitrogen-containing compounds. Therefore, the experimental design included both synthetic procedures and a combination of characterization techniques to elucidate the structure of the modifying agents and to determine the morphological characteristics and the adsorption properties of the silica materials.

### 3.1. Synthesis of SAPTES

SAPTES presents a Schiff base which was derived from furfural and APTES under mild reaction conditions, i.e., stirring of the reagents in an equimolar ratio at room temperature for 24 h (Figure 1). It was a high-yield reaction, and a complete conversion was achieved.

The ^1^H NMR spectrum (Figure 2a) of SAPTES showed a signal at 8.0 ppm which is assigned to the H-atom in the formed aldimine bond. The ^13^C{H} NMR spectrum (Figure 2b) also provided evidence for the formation of a Schiff base—this is the signal at 151.58 ppm (–CH=N) and the lack of a signal at about 178 ppm for the aldehyde carbon atom in starting furfural. The signals due to the furanyl ring and the propyltriethoxysilane segment are also seen in the ^1^H and ^13^C{H} NMR spectra (Figure 2a and Appendix A), and their assignments are given in the experimental section. The obtained data and assignments are consistent with the published data for the starting reagents and aldimine structures [31,32,33]. ^1^H NMR data concerning the ratio of the integral intensities of the signals accounting for the number of the protons in the corresponding structure evidenced that the reaction proceeded quantitatively and there was no need for a purification step.

The IR spectral data of SAPTES also confirmed the structure of the synthesized Schiff base. The absorption band at 1643 cm^−1^ is a characteristic one for the CH=N stretching vibrations (Figure 3). The strong absorptions in the spectral region 1100–1014 cm^−1^ are attributed to C–O–C stretching in the furanyl ring and Si-OCH_2_ stretching in the triethoxysiicate moiety [26]. A band due to the C-H deformations of the furanyl ring is also seen at 750 cm^−1^ [34].

### 3.2. Synthesis of PAPTES

The reaction between the diesters of H-phosphonic acid and imine derivatives is known to yield aminophosphonates [35]. The addition of the P–H group to the azomethine bond could proceed without a catalyst at temperatures above 80 °C and for long reaction times. In the presence of a catalyst, i.e., sodium alcoholate or cadmium iodide, the reaction could be carried out at lower temperatures, achieving yields above 90% [25,26].

In the present study, a modifying agent containing a polar phosphoryl group was obtained via the reaction of SAPTES with diethyl ester of the H-phosphonic acid (Figure 2). The conversion of the reaction in absence of a catalyst was low (about 40%); therefore, cadmium iodide was used as the catalyst. Anhydrous diethylene glycol dimethyl ether (diglyme) was used as a reaction medium, being a good solvent for the reactants and the catalyst. The addition of the H-phosphonate to the Schiff base proceeded quantitatively, and the obtained aminophosphonate derivative of APTES, denoted as PAPTES, was confirmed by IR and NMR spectroscopy. The characteristic band for the azomethine group at 1643 cm^−1^ disappeared, and a new absorption at 1250 cm^−1^ was observed. The latter is characteristic of the phosphoryl group (ν(P=O)). The strong absorptions in the spectral region 1103–1024 cm^−1^ attributed to C–O–C and Si–OCH_2_ stretching were enhanced due to the contribution of the P-OCH_2_ stretching from the phosphonate segment (Figure 3). These features in the IR spectrum of PAPTES evidenced the formation of the desired product. The comparison of the ^1^H NMR spectra of SAPTES (Figure 2a) and that of the addition product (Figure 4a) confirmed the yield of the latter. The signal for the hydrogen from the imine –N=CH-group at 8.00 ppm disappeared, and a multiplet between 3.87 ppm and 3.83 ppm was seen as proof of the formation of the -HNCH(Fur)P(O)- motif (see also Appendix A). Another change is the up-field shift of the signal for the hydrogens from methylene group adjacent to the nitrogen atom: from the range of 3.52–3.49 ppm (–CH_2_N=CH–) to the range of 2.53–2.40 ppm (–CH_2_NHCH–). The distribution of the electron density in the furan ring was also changed, and this reflected in the chemical shifts of the signals for the hydrogens in the furanyl residue. The signals between 3.72 and 3.25 ppm belong to the solvent used.

The ^13^C NMR spectrum of PAPTES (Appendix A) also provided data that support the formation of the aminophosphonate derivative. The assignment of the signals is presented in the experimental section. Correspondingly, the signal at 151.58 ppm for the C-atom in the azomethine bond was not present, and a dublet appeared at 54.61 ppm, with a coupling constant of ^1^J = 160.4 Hz. This signal was attributed to the C-atom bound to the P-atom in the formed –HN**C**HP(O)– segment. The addition of the phosphonate structure in the molecule led to the splitting of other signals into doublets—the signals of the furanyl ring and the methylene group next to the amino group (–CH_2_NH–). The doublets of the CH_3_CH_2_OP(O)– residues are also present in the spectrum at about 63 ppm and 16 ppm for the methylene and methyl carbons, respectively. The ^31^P{H} NMR spectrum of PAPTES (Figure 4b) shows a singlet at 21.12 ppm, i.e., in the spectral region expected for the aminophosphonate derivatives. In the ^31^P NMR spectrum, this signal presents a sextet, which serves as additional proof for the structure of the new product, as presented in Figure 2. A second signal at 7.27 ppm is also observed, and it is a dublet of triplets. It was attributed to unreacted diethyl H-phosphonate in a quantity of 4.76 mol.%.

### 3.3. Modification of MCM-48 and SBA-15

For the modification of MCM-48 and SBA-15, SAPTES and PAPTES were used as modifying agents (Figure 3).

All samples show similar features of the IR spectra of silica materials, i.e., the strong absorption in the spectral region 1100–950 cm^−1^ attributed to the Si-O-Si stretching vibrations of the silica matrix. In Figure 3, the IR spectra of the original MCM-48 and the modified samples MCM-48/SAPTES and MCM-48/PAPTES are presented. Besides the dominating absorption of the silica matrix, the presence of a band at 1643 cm^−1^ due to the azomethine group in the IR spectrum of MCM-48/SAPTES is also seen. Correspondingly, in the IR spectrum of MCM-48/PAPTES, a shoulder at 1235 cm^−1^ is observed, resulting from the grafted phosphonate moieties. IR spectra of SBA-15 and its modified sample displayed similar spectral features (Appendix A). The spectral data evidenced the introduction of new functional groups onto the porous surface of the silica materials.

The incorporation of organic moieties SAPTES and PAPTES in MCM-48 was further evidenced by solid-state ^1^H → ^29^Si and ^1^H → ^13^C CP MAS NMR spectroscopy. Appendix A shows the ^1^H → ^29^Si CP MAS spectra of MCM-48 silica materials functionalized with SAPTES and PAPTES (Appendix A), showing the typical resonances at −109 and −101 ppm for Q4 and Q3 Si structural units, representing the main building blocks of the silicate framework [Q^n^ = Si(OSi)_n_(OH)_4−n_, n = 2–4], while the signals at −66 and −58 ppm are characteristic for organosiloxane structural fragments T^3^ [(SiO)_3_Si–R] and T^2^ [(SiO)_2_Si–(R_1_)–OR_2_], respectively. Appendix A shows the ^31^P NMR spectrum of PAPTES, where the intense resonance at 21.5 ppm originates from the phosphorus atom of the linked organic segment (PAPTES), while the small signal at 6.9 ppm is from the unreacted diethyl phosphonate.

The ^1^H → ^13^C CP MAS NMR spectra show the characteristic resonances of the structural fragments of PAPTES and SAPTES (Figure 5a and Appendix A, respectively). The assignment of the chemical shifts for MCM-48 functionalized with SAPTES are as follows: 151 ppm (=C–O furan ring, and Si–CH_2_–CH_2_–CH_2_N=CH–), 146 ppm (=CH–O furan ring), 111 ppm (CH–CH, furan ring), 63 ppm (Si–CH_2_–CH_2_–CH_2_N–), 23 ppm (Si–CH_2_–CH_2_–CH_2_N–) and 9 ppm (Si–CH_2_–CH_2_–CH_2_N). The chemical shifts for PAPTES-functionalized MCM-48 materials are as follows: 149 ppm (=C–O furan ring), 143 ppm (=CH–O furan ring), 111 ppm (CH–CH, furan ring), 63 ppm (Si–CH_2_–CH_2_–CH_2_N– and PO–CH_2_–CH_3_), 53 ppm (Si–CH_2_–CH_2_–CH_2_N–CH–P), 23 ppm (Si–CH_2_–CH_2_–CH_2_N–), 18 ppm (PO–CH_2_–CH_3_) and 9 ppm (Si–CH_2_–CH_2_–CH_2_N).

### 3.4. Thermogravimetric Analysis (TGA)

The loadings of SAPTES and PAPTES in the SBA-15 and MCM-48 were calculated by using TGA analysis (Figure 6). For SBA-15/PAPTES and SBA-15/SAPTES, the <10% weight loss below 140 °C might be due to the removal of the physically and chemically adsorbed water. Additionally, the intensive peak MCM-48/PAPTES at 104 °C is probably due to the release of retained water and solvent in the tridimensional and narrower pore system of MCM-48. For all modified samples, the weight loss above 140 °C is attributed to the decomposition of the grafted organic residues (Figure 6). The decomposition of PAPTES in SBA-15/PAPTES and MCM-48/PAPTES silicas occurs in the temperature interval of 182–188 °C, which is significantly lower than that needed for SAPTES decomposition from SBA-15/SAPTES and MCM-48/SAPTES materials (345–355 °C, 485 °C). The effect of the structure for the PAPTES decomposition is negligible, showing that a 6 °C higher temperature is needed for PAPTES decomposition in SBA-15 in comparison to that needed for its MCM-48 supported analog. However, the SAPTES decomposition in MCM-48/SAPTES occurs in two steps (345 °C and 485 °C), whereas, for SBA-15/SAPTES, only one step is registered at 355 °C. This effect could be explained by the more open structure and bigger pores of SBA-15 than those of MCM-48. The weight changes above 550 °C are related to the structural changes of the silica supports, due to dehydroxilation processes.

The TG analysis of the SAPTES-modified samples shows a weight loss of 19.07 wt.% for MCM-48 and 23.25 wt.% for SBA-15. Significantly higher weight loss is registered in the TG experiments for PAPTES-modified MCM-48 and SBA-15, 40.20 wt.% and 28.85 wt.%, respectively (Table 1). The greater amount of PAPTES incorporated in the mesoporous silicas could be explained by its higher molar mas and bulk molecule compared to those of SAPTES. Additionally, the peculiarities of the three-dimensional matrix of MCM-48 also contribute to the high extent of modification in the case of MCM-48/PAPTES.

### 3.5. Physicochemical Characterization of the Modified MCM-48 and SBA-15 Materials

The nitrogen adsorption–desorption isotherms of the initial and the modified MCM-48 and SBA-15 are presented in Figure 7. The isotherms of the parent and the modified MCM-48 exhibit a sharp increase at a relative pressure between p/p_o_= 0.2–0.4, which is associated with the capillary condensation of nitrogen in the channels and also an indication of narrow pore size distribution (Figure 7a). The isotherms of the MCM-48 samples are reversible and do not show any hysteresis loop. The modified samples are characterized by a somewhat lower specific surface area and decreased pore diameter and total pore volume.

The isotherms of the SBA-15 samples are of type IV, with a hysteresis loop at 0.6–0.7 relative pressure, typical for a SBA-15 structure (Figure 7b). A decrease of the textural parameters, such as surface area and total pore volume of the modified samples, was registered. Physicochemical and structural parameters of the samples obtained by N_2_ isotherms are summarized in Table 1. The nitrogen physisorption measurements of SBA-15 show a surface area of 926 m^2^/g and pore volume of 1.22 cm^3^/g. The MCM-48 material shows a much higher surface area (1235 m^2^/g) and pore volume (0.83 cm^3^/g). A significant decrease in the S_BET_ and pore volume was observed after modification of SBA-15 and MCM-48 with both modifying agents. The SBA-15/PAPTES showed a higher decrease in surface area and pore volume than that in the case of SAPTES. Probably, this is due to the bulkier structure of PAPTES compared to SAPTES. MCM-48/PAPTES displayed textural parameters similar to those of MCM-48/SAPTES. The strong effect of the post-synthetic treatment on the textural parameters of both MCM-48 modifications was due to the partial deterioration of the three-dimensional structure of the MCM-48 silica with small pore sizes around 2.4 nm during the modification procedure. The significant decrease of the surface area and pore volume of MCM-48/SAPTES and MCM-48/PAPTES is probably due to the total pore blocking of some pores with grafted groups.

Low-angle XRD data of the initial SBA-15 and MCM-48 samples confirm the formation of the hexagonal and cubic mesoporous structure, respectively. The functionalization of SBA-15 and MCM-48 resulted in decreased intensity and some broadened reflections, indicating some structural disorder (Appendix A). These observations are typical for functionalized mesoporous silicas.

However, the partial deterioration of the MCM-48 structure leads to an increase of the pore diameter of the still-not-blocked pores, and this is in good accordance with the changes of the other structural parameters, assuming a small decrease in the structure ordering for MCM-48. The bigger pores and the more open structure of SBA-15 are the reasons for the preservation of the mesoporous structure during the modification by SAPTES and PAPTES. The decrease of surface area, pore volume and pore diameter for the SBA-15/SAPTES and SBA-15/PAPTES silicas were also detected.

The UV–Vis spectra of the modified samples are presented in Figure 8. The modification with SAPTES of SBA-15 and MCM-48 leads to the appearance of an intensive peak at 270 nm and a broad shoulder in the visible spectral region, with the latter being more intensive for MCM-48/SAPTES. UV–Vis spectra of the MCM-48/PAPTES and SBA-15/PAPTES show the presence of peaks at 230 and 270 nm. These spectral features correspond to the UV absorptions of the two modifying agents shown in Appendix A. The absence of intensive absorptions in the visible region for the PAPTES functionalized matrices is an expected observation, due to the addition of the phosphonate moiety to the azomethine group and lack of the conjugated segment of SAPTES.

### 3.6. CO_2_ Adsorption

SBA-15 and MCM-48 modified with SAPTES and PAPTES were tested as adsorbents for CO_2_ capture under dynamic conditions. Breakthrough curves for CO_2_ adsorption in dynamic conditions with 3% CO_2_/N_2_ flow are shown in Figure 9.

The adsorption capacities for CO_2_ under dynamic conditions (3 mL/min at a nitrogen flow of 27 mL/min, atmospheric pressure, 25 °C) were calculated, and the results are presented in Table 2. It was found that the modified mesoporous silicas adsorbed a higher amount of CO_2_ than the initial ones. The adsorption behavior of the initial MCM-48 and SBA-15 materials is influenced by the presence of silanol groups, which are the main adsorption sites. The presence of smaller amounts of silanols in SBA-15 leads to its lower adsorption capacity. Moreover, the period needed for achieving the total adsorption capacity for the porous structure of MCM-48 (T = 23 min) (Figure 9a–c) is longer than that needed for the SBA-15 silica (T = 14 min) (Figure 9d–f). This result indicates that the interpenetrating network of the three-dimensional pores of MCM-48 retards the access to some adsorption sites in comparison to the two-dimensional pores in the hexagonal SBA-15. The modification of MCM-48 leads to an increase in the capacity of the adsorbent by about 2 times, and values of 4.2–4.6 mmol/g have been achieved. The significantly higher adsorption capacity found for the modified MCM-48 materials is due to the peculiarities of the carrier structure—a favorable combination of a high specific surface area and a three-dimensional structure. The experiments for CO_2_ desorption were performed, and the total desorption was determined at 40 °C in 20 min. The adsorption of CO_2_ was recorded in three adsorption cycles that showed a similar adsorption performance for the initial and the modified silicas (Table 2). In Table 2, we can see the CO_2_ adsorption uptake increase in the following order: SBA-15 < MCM-48 ≈ SBA-15/SAPTES < SBA-15/PAPTES < MCM-48/PAPTES < MCM-48/SAPTES.

The functionalization of the adsorbent pore surface strongly enhances the effectiveness of CO_2_ adsorption, specifically at low CO_2_ pressure. It was reported that the presence of electron-rich units and high charge density at N-sites facilitated the uptake of polarizable CO_2_ molecules through local dipole–quadrupole interactions [36]. Yaseen et al. investigated Schiff bases derived from trimethoprim and aromatic aldehydes as adsorbents for carbon dioxide capture. Their high CO_2_ adsorption capacity was attributed to the appropriate pore size distribution and the polar nature of the heteroatoms (N and O), as well as the Lewis-base nature of the new materials [37]. It was shown that polymers containing aromatic and phosphate units were suitable for CO_2_ capture. The interaction between polarized groups and gas molecules contributed to the CO_2_ uptake, which was 14 wt.% at 50 bars [38]. In the present work, the modified silica materials afford a combination of factors, including a mesoporous structure and polar sites to strongly interact with CO_2_. SAPTES possesses electron-rich moieties—the aldimine group was conjugated to the furanyl ring, while PAPTES contains a secondary amino group and a highly polar phosphoryl (P=O) group, in addition to the furanyl ring. The gas uptake capacity of the functionalized adsorbents can be explained with favorable interactions between the polarizable CO_2_ molecules and the polar sites on the pore surface, including Lewis-base/Lewis-acid interactions between the electron-rich functions and CO_2_ molecules.

In regard to the above discussion, the nature of the adsorbed CO_2_ was elucidated by using two different types of NMR experiments [14,39]. We used NMR experiments with cross-polarization from protons to neighboring carbons (^1^H → ^13^C cross-polarization (CP)) to detect the presence of chemisorbed ^13^CO_2_. It is expected that, in the CP spectra, the signal of chemisorbed ^13^CO_2_ will be selectively enhanced, due to transfer of magnetization from protons of the organic structural fragments to the carbon atom from the chemically adsorbed ^13^CO_2_ species. According to the literature data for aminomodified mesoporous materials, the resonances of the chemisorbed CO_2_ species are expected to be in the chemical shift range of 159–167 ppm [14,39]. The CP experiments are not suitable for registration of physically adsorbed ^13^CO_2_, due to inefficient CP transfer, as a result of its high mobility. Therefore, ^13^C spectra with high-power proton decoupling (HPDEC) were measured to register the physically adsorbed ^13^CO_2_. The ^1^H → ^13^C CP spectra of CO_2_-loaded materials do not show additional signals that are characteristic for chemisorbed CO_2_ species (Figure 5b and Appendix A) as compared with the spectra of the parent materials (Figure 5a and Appendix A), while the ^13^C HPDEC spectra (Figure 5c and Appendix A) indicate the presence of physically captured CO_2_, which is evidenced by the additional signals at 128 ppm.

Considering the high CO_2_ adsorption capacities of the SAPTES- and PAPTES-modified samples, it is important to compare their capacities with that of the other types of the modified MCM-48 and SBA-15 materials [13]. Moreover, 1-methylpiperazine-modified mesoporous MCM-48 showed similar adsorption capacities for CO_2_ (4.4 mmol/g) and proved chemisorption of the gas. MCM-48 grafted with PAPTES or SAPTES residues displayed similar values of CO_2_ adsorption capacities, though the CO_2_ capture was due to the physisorption process. The combined experimental data underline the significant effect of the matrix structural characteristics, as well as the possibility to enhance the adsorbent performance via the appropriate modification. The grafting of PAPTES or SAPTES onto the silica surface afforded polar adsorption sites and changes in the textural characteristics that synergistically increased the CO_2_ adsorption capacities of the silica material to levels obtained for modifications with amines. In certain applications, the lower temperatures needed for CO_2_ desorption (40 °C) could be an advantage.

In Table 3, the comparison of our new data with those that are already published is presented. Despite the applied atmospheric CO_2_ pressure in the adsorption experiments, the determined capacity of our SAPTES- and PAPTES-modified MCM-48 silicas is higher than that of other N-modified mesoporous silica and zeolite materials, with the exception of N-(2-aminoethyl)-3-aminopropyl/SBA-16, which was determined at 30 bars.

The heats of adsorption of all samples were calculated from the CO_2_ adsorption isotherm at 25 and 0 °C, using the Clausius–Clapeyron equation. The heat of adsorption values of the samples ranged between 1 and 30 kJ/mol.

SAPTES- and PAPTES-functionalized SBA-15 silicas exhibited a steep decrease in the heats of adsorption as a function of the adsorbed amount of CO_2_ in comparison with the parent SBA-15 silica. This effect is related to the heterogeneity of the absorption sites. It could be assumed that the silanol groups in the micropores of SBA-15 are not affected by the modification process and participate in the CO_2_ adsorption together with the grafted functional groups. However, such effect is not observed for MCM-48 varieties. That implies relative homogeneous distribution of the modified adsorption sites on the matrix pore surface. We could note that the heat of adsorption determined for the SAPTES-modified materials is higher than that measured for the PAPTES-modified ones (Figure 10). It presumes a stronger interaction between the CO_2_ molecules and the grafted Schiff base residues. In our previous paper, the modification by 1-methylpiperazine and morpholine of MCM-48 and SBA-15 resulted in the higher heats of adsorption (40–50 kJ/mol) due to the stronger interaction between functional groups and CO_2_ molecules. We also determined a higher temperature for CO_2_ desorption from 1-methylpiperazin-MCM-48 (60 °C) than for SAPTES and PAPTES—modified silicas (40 °C) which could have a beneficial effect on their application.

However, the CO_2_ adsorption capacities achieved with the PAPTES-modified adsorbents are similar to that of the SAPTES grafted materials. The partial pore-blocking effect of the PAPTES in the modified samples resulted in the restricted diffusion of CO_2_ molecules and its effective capture in the pore volume of the matrix. The obtained results are a consequence of a balance between the matrix textural parameters and the functional and structural features of the modifying agents.

## 4. Conclusions

Novel (3-aminopropyl)triethoxysilane derivatives containing azomethine bond conjugated with furanyl ring (SAPTES) or aminophosphonate segment (PAPTES) were prepared and applied in the post-synthesis modification of MCM-48 and SBA-15 silicas. The textural characterization of the obtained modified materials showed the preservation of the mesoporous structure during the modification procedure. A higher capacity for CO_2_ adsorption was determined for all modified samples in comparison to the initial ones. The SAPTES-modified MCM-48 silica showed the highest adsorption capacity of 4.60 mmol/g for CO_2_ adsorption. The modified silicas kept their adsorption ability in three adsorption cycles. CO_2_ was physically retained on the sorbent surface, as evidenced by a solid-state ^13^C NMR technique. The total CO_2_ desorption was achieved at 40 °C. Adsorbents with such parameters could be considered in certain applications, due to the high adsorption capacity and the low energy consumption for CO_2_ desorption.

## Data Availability

The data presented in this study are available on request from the corresponding author.

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
