# Peer review of "CO2 Adsorption on the N- and P-Modified Mesoporous Silicas"

_nanomaterials, 2022, doi:10.3390/nano12071224_

Round 1

Reviewer 1 Report

In this work, the adsorption of CO2 on the modified silica was studied using experimental methods. The manuscript may be accepted for publication after some minor additions/corrections.

1) Introduction:–  Molecular simulation is also a useful tool  to understand the adsorption of CO2 on different substrates  (cite, e.g., https://doi.org/10.1021/acsearthspacechem.1c00160).

2) Fig. 9.: Provide comparison for MCM and SBA with experimental data from previous studies, if available.

3) p.14, line 389:”heat adsorption” – heat of adsorption

Author Response

The authors thank the referees for the remarks and suggestions. The needed corrections have been made. The corrected version of the manuscript has been submitted.

Reviewer 1

In this work, the adsorption of CO2 on the modified silica was studied using experimental methods. The manuscript may be accepted for publication after some minor additions/ corrections.

Remark: Introduction:  Molecular simulation is also a useful tool to understand the adsorption of CO2 on different substrates (cite, e.g., ttps://doi.org/10.1021/acsearthspacechem.1c00160).

Answer: A paragraph dedicated to the molecular simulations as useful tool for evaluation of the effects and CO2 adsorption has been added with the suggested reference, and one more, in the introduction part (p. 2).

The molecular simulation studies have been applied to evaluate the effect of different factors on the adsorbent CO2 capture performance. The computationally explored parameters include surface defects, temperature, humidity, adsorbate composition, etc. The results of the simulation studies provide comprehensive understanding of the surface interactions and assistance in the design of the adsorbent material

  • C. Kim, S. S. Jang. Molecular Simulation Study on Factors Affecting Carbon Dioxide Adsorption on Amorphous Silica Surfaces. J. Phys. Chem. C 2020, 124, 23, 12580–12588. https://doi.org/10.1021/acs.jpcc.0c03035;
  • N. Nair, R. Cui, and S. Sun. Overview of the Adsorption and Transport Properties of Water, Ions, Carbon Dioxide, and Methane in Swelling Clays. ACS Earth and Space Chemistry 2021, 5, 10, 2599-2611. https://doi.org/10.1021/acsearthspacechem.1c00160 .

Remark: Fig. 9: Provide comparison for MCM and SBA with experimental data from previous studies, if available.

Answer: The comparison of our results with the already published ones in our previous paper has been made and added in the manuscript on p.17-18. A table (Table 3) has been provided with comparative data.

Remark: p.14, line 389:”heat adsorption” – heat of adsorption

Answer: The needed correction was made. We appologise for the mistake.

Reviewer 2 Report

Manuscript title : CO2 Adsorption on the N- and P-Modified Mesoporous Silicas

In this work, mesoporous silicas bansed on SBA-15 are synthetized and modified via the chemical grafting of APTES and SAPTES. The paper represents some interesting results on CO2 adsorption, and the manuscript need major revision before possible publication:

  • In the abstract, the most interesting results should be added to show the originality of the work, like the surface properties, comparison between the prepared materials in term of CO2 adsorption capacity.
  • In the abstract : triethoxysilanes should be replaced by (3-aminopropyl)triethoxysilane) and its abbreviation (APTES)
  • Line 14: The physicochemical characteristics of the adsorbent samples were 14 studied by nitrogen physisorption and UV-Vis spectroscopy. It will be better add the type of the physicochemical properties and remove (The physicochemical characteristics).
  • In the introduction: authors could discuss more works aimed in APTES utilization fo CO2 adsorption, and SBA15, bellow are some references that can be used to discuss and compare the obtained results:
  1. https://doi.org/10.1016/j.micromeso.2018.03.024
  2. https://doi.org/10.1016/j.molstruc.2019.04.035
  3. https://doi.org/10.1016/j.cej.2018.02.084
  4. https://doi.org/10.1016/j.apsusc.2010.04.066
  • The experimental part: Please change ml to mL
  • I see that many steps are used for the preparation and functionalization of the materials, can the authors illustrate a scheme to summarize the synthesis route including all steps.
  • Authors could merge the 2.7.1 to 2.7.5 into two part 2.7.1: Materials characterization and 2.7.2: CO2 adsorption measurements
  • Results and discussions: The weaknesses in this part reside in the discussion of the nature of adsorption of CO2, authors need more interpretation to explain the adsorption phenomenon over unmodified and modified materials.
  • Authors could use the physicochemical characterization to confirm the results obtained for CO2 adsorption and then, conclude about the CO2 interaction. Here we need more attentions to explain surface structure and th involved interaction between CO2 and functional groups.
  • To show the novelty, authors could compare the obtained results with other work published previously

Author Response

The authors thank the referees for the remarks and suggestions. The corrections needed have been made. The corrected version of the manuscript has been submitted.

Reviewer 2

In this work, mesoporous silicas based on SBA-15 are synthetised and modified via the chemical grafting of APTES and SAPTES. The paper represents some interesting results on CO2 adsorption, and the manuscript needs major revision before possible publication:

Remark: In the abstract, the most interesting results should be added to show the originality of the work, like the surface properties, comparison between the prepared materials in term of CO2 adsorption capacity.

Answer: Thank you for the suggestion. The abstract has been revised and additional data obtained in this study were added. Results from additionally performed solid state NMR experiments (13C NMR spectra with high power proton decoupling) that proved the physical nature of the interaction of CO2 with the functionalised support, and data about the total CO2 desorption.

Remark: In the abstract : triethoxysilanes should be replaced by (3-aminopropyl) triethoxysilane) and its abbreviation (APTES)

Answer: The needed correction was made.

Remark: Line 14: The physicochemical characteristics of the adsorbent samples were studied by nitrogen physisorption and UV-Vis spectroscopy. It will be better add the type of the physicochemical properties and remove (The physicochemical characteristics).

Answer: The needed correction has been made.

Remark: In the introduction: authors could discuss more works aimed in APTES utilization for CO2 adsorption, and SBA15, bellow are some references that can be used to discuss and compare the obtained results:

  1. https://doi.org/10.1016/j.micromeso.2018.03.024
  2. https://doi.org/10.1016/j.molstruc.2019.04.035
  3. https://doi.org/10.1016/j.cej.2018.02.084
  4. https://doi.org/10.1016/j.apsusc.2010.04.066

Answer: A paragraph containing a brief overview of APT(M)ES use in adsorbent functionalization to enhance CO2 adsorption was introduced in the Introduction section. The mentioned papers have been added in the reference list and discussed.

Similarly, triamine-functionalised SBA-15 had a greater CO2 retention capacity value than the grafted with mono- and di-amines matrices. The presence of moisture enhanced the adsorbent performance suggesting the participation of water molecules in CO2 retention [ref 1]. 3-Aminopropyltri(m)ethoxisilanes (APMES or APTES) were used for functionalisation of plant-derived sorbents [ref 2, ref.3]. Mesoporous SBA-15 synthesized from rice husk ash was grafted with APTMS for subsequent growth of tris(2-aminoethyl) amine (TREN) dendrimers. A CO2 adsorption capacity of 5–6 and 7–8 wt% was observed for the material with second and third dendrimer generation which was higher than the reported values for melamine and PAMAM dendrimers [ref.3]. APTES was used for modification of cocoa shell based hydrochar (HC) followed by cobalt particle incorporation.  The obtained HC-APTES-Co sorbent displayed higher CO2 retention capacity than its metal-free counterpart, in spite of its lower basicity and porosity. The material afforded high surface affinity towards CO2 though the direct contact between CO2 and the amino groups was hampered. The achieved result was reversible capture of CO2 with easy regeneration due to predominant physical CO2 condensation [ref 2].

Ref 1: B. Boukoussa, A. Hakiki, N. Bouazizi, A.-P. Beltrao-Nunes, F. Launay, A. Pailleret, F. Pillier, A. Bengueddach, R. Hamacha, A. Azzouz. Mesoporous silica supported amine and amine-copper complex for CO2 adsorption: Detailed reaction mechanism of hydrophilic character and CO2 retention. J. Molecular Structure, Vol. 1191, 2019, 175-182].

Ref. 2: J. Vieillard, N. Bouazizi, R. Bargougui, N. Brun, P. Fotsing Nkuigue, E. Oliviero, O. Thoumire, N. Couvrat, E. Djoufac Woumfo, G. Ladam, N. Mofaddel, A. Azzouz, F. Le Derf. Cocoa shell-deriving hydrochar modified through aminosilane grafting and cobalt particle dispersion as potential carbon dioxide adsorbent. Chem. Eng. J. Volume 342, 2018, 420-428].

Ref. 3: M. Bhagiyalakshmi, S. Do Park, W. S. Cha, H. T. Jang. Development of TREN dendrimers over mesoporous SBA-15 for CO2 adsorption. Applied Surface Science 256 (2010) 6660–6666. doi:10.1016/j.apsusc.2010.04.066;

Remark: The experimental part: Please change ml to mL

Answer: We corrected.

Remark: I see that many steps are used for the preparation and functionalization of the materials, can the authors illustrate a scheme to summarize the synthesis route including all steps.

Answer: A scheme that presents the steps of the experimental design was prepared and added in the experimental part (Figure 1).

Remark: Authors could merge the 2.7.1 to 2.7.5 into two part 2.7.1: Materials characterization and 2.7.2: CO2 adsorption measurements

Answer: The advice was followed and the corrections introduced.

Remark: Results and discussions: The weaknesses in this part reside in the discussion of the nature of adsorption of CO2, authors need more interpretation to explain the adsorption phenomenon over unmodified and modified materials.

Answer: The discussion was extended with additional interpretation of the CO2 results and examples from literature materials with similar functions (on p. 17). In regard to the above said, the nature of the adsorbed CO2 was elucidated by using two different types of NMR experiments: the one with cross polarization from protons to neighbouring carbons is applicable to detect the presence of chemisorbed 13CO2. It is expected  that in the CP spectra the signal of chemisorbed 13CO2 in the range 159-167 ppm to be detected due to selectively enhanced  transfer of magnetization from protons of the organic structural fragments to the carbon atom from the chemically adsorbed 13CO2 species. On the other hand, the CP experiments are not suitable for registration of physically adsorbed 13CO2, due to inefficient CP transfer as a result of its high mobility. Such signals were not observed and therefore 13C spectra with high power proton decoupling (HPDEC) were measured to register the physically adsorbed 13CO2 and they indicate the presence of physically captured CO2 evidenced by the additional signals at 128 ppm. New figures were provided (Figures 5, Figure S5 and Figure S6) and corresponding discussion on the new data.

Remark: Authors could use the physicochemical characterization to confirm the results obtained for CO2 adsorption and then, conclude about the CO2 interaction. Here we need more attentions to explain surface structure and thе involved interaction between CO2 and functional groups.

Answer: Please, see the answer to the remark above.

Remark: To show the novelty, authors could compare the obtained results with other work published previously

Answer: The comparison of our results with the already published ones was extended. The data are summarized in Table 3 (added on page 17-18).

Reviewer 3 Report

The manuscript by Tumurbaatar et al. deals with CO2 Adsorption on N- and P-Modified Mesoporous Silicas. It is an interesting work that may be published upon revision. I have the following comments:

Labeled (numbered) atoms in presented NMR results (i.e. lines 113-119) should be also labeled in relevant structures and figures. It would be better if you labeled each atom in the structure, add it as an inset in the free (white) space in figures as in Fig 1a and Fig 1b, and in addition, labeled all peaks in the figures with the same numbers.

NMR and IR results in lines 114-120. I haven’t seen before the division symbol ÷ between numbers (NMR) or two different peaks (IR). What is the physical meaning for this?

You should present as SI zoom in regions of Fig 1a (0.5-4.0 ppm) and Fig 3a (0.4-4.5 ppm).

Fig 4, it is obvious that you have altered spectrum upon modification with PAPTES, but I really don’t see anything different with SAPTES or for the case of SBA in Fig S2. Moreover, it would be better if you also presented in the same figure IR spectra of SAPTES and PAPTES. Line 276: it is figure 4 and not 5. Also correct CMC to MCM in figure.

TG analysis: I am not sure you can support that chemically adsorbed water can be removed below 100 C. I would say it is 100% physically adsorbed on the porous materials. Weight loss is very gradual, and you don’t present in advance the DTA curves. Even though the organic moieties you grafted have many different groups, I would expect to see more define regions with weight loss accompanied by relevant exothermic peaks.

Fig 6 and 7 quality is very low.

You should add UV-Vis spectra of pure SAPTES and PAPTES in Fig 7 and explain better the two peaks at 230 and 270 nm.

XRD of the materials is vital for their structural characterization and is missing. Furthermore, as I can understand from IR spectra and nitrogen adsorption/desorption isotherms, materials don’t possess high crystallinity.

Even though there are some references in the introduction, results of table 2 should be discussed in regard with data from bibliography for relevant materials in the results and discussion section.

Fig 9, again very low-quality figure and try to use totally different colors for depicting curves.

Author Response

The authors thank the referees for the remarks and suggestions. The corrections needed have been made. The corrected version of the manuscript has been submitted.

Reviewer 3

The manuscript by Tumurbaatar et al. deals with CO2 Adsorption on N- and P-Modified Mesoporous Silicas. It is an interesting work that may be published upon revision. I have the following comments:

Remark: Labeled (numbered) atoms in presented NMR results (i.e. lines 113-119) should be also labeled in relevant structures and figures. It would be better if you labeled each atom in the structure, add it as an inset in the free (white) space in figures as in Fig 1a and Fig 1b, and in addition, labeled all peaks in the figures with the same numbers.

Answer: The advice of the reviewer was followed and product structures with labelled atoms were added in Fig. 1a and Fig. 1b.

Remark: NMR and IR results in lines 114-120. I haven’t seen before the division symbol ÷ between numbers (NMR) or two different peaks (IR). What is the physical meaning for this?

You should present as SI zoom in regions of Fig 1a (0.5-4.0 ppm) and Fig 3a (0.4-4.5 ppm).

Answer: The symbol ÷ was used in meaning “from …. to ….” The symbol was replaced with a dash.

 Zoomed spectral regions (0.5-4.0 ppm) (0.4-4.5 ppm) from Fig. 1a and Fig. 3a, respectively, were presented in the SI. New figures: Figure S1 and Figure S2.

Remark: Fig 4, it is obvious that you have altered spectrum upon modification with PAPTES, but I really don’t see anything different with SAPTES or for the case of SBA in Fig S2. Moreover, it would be better if you also presented in the same figure IR spectra of SAPTES and PAPTES. Line 276: it is figure 4 and not 5. Also correct CMC to MCM in figure.

Answer: The authors followed the advice and the spectral data were reorganized in the prepared figures Figure 3a and b; and Figure S4. The typing error was also corrected. Please, accept our excuse.

Remark: TG analysis: I am not sure you can support that chemically adsorbed water can be removed below 100 C. I would say it is 100% physically adsorbed on the porous materials. Weight loss is very gradual, and you don’t present in advance the DTA curves. Even though the organic moieties you grafted have many different groups, I would expect to see more define regions with weight loss accompanied by relevant exothermic peaks.

Answer: We agree with the reviewer’s remarks and the paragraph dedicated to the TGA analysis was revised as follows: TGA analysis (the previous figure was extended with a dtg curve: Figure 6). For SBA-15/PAPTES and SBA-15/SAPTES, the <10 % weight loss below 140 °C might be due to the removal of the physically and chemically adsorbed water. Additionally, the intensive peak MCM-48/PAPTES at 104°C is probably due to the release of retained water and solvent in the tridimensional and narrower pore system of MCM-48. For all modified samples, the weight loss above 140 °C is attributed to the decomposition of the grafted organic residues. The decomposition of PAPTES in SBA-15/PAPTES and MCM-48/PAPTES silicas occurs in the temperature interval 182-188°C, which is significantly lower than that needed for SAPTES decomposition from SBA-15/SAPTES and MCM-48/SAPTES materials (345-355°C, 485 °C). The effect of the structure for the PAPTES decomposition is negligible, showing that 6°C higher temperature is needed for PAPTES decomposition in SBA-15 in comparison to that needed for its MCM-48 supported analog. However, the SAPTES decomposition in MCM-48/SAPTES occurs in two steps (345 °C, 485 °C) whereas for SBA-15/SAPTES only one step is registered at 355°C. This effect could be explained by the more open structure and bigger pores of SBA-15 than those of MCM-48. The weight changes above 550°C are related to the structural changes of the silica supports due to dehydroxilation processes.”  

Remark: Fig 6 and 7 quality is very low.

Answer: The figures’ quality has been improved.

Remark: You should add UV-Vis spectra of pure SAPTES and PAPTES in Fig 7 and explain better the two peaks at 230 and 270 nm.

Answer: The UV-Vis spectra of pure SAPTES and PAPTES were added in the SI. The two peaks at 230 and 270 nm are due to UV absorbance by the functionalities of the grafted SAPTES and PAPTES mieties.

Remark: XRD of the materials is vital for their structural characterization and is missing. Furthermore, as I can understand from IR spectra and nitrogen adsorption/desorption isotherms, materials don’t possess high crystallynity.

Answer: The low angle XRD are presented in Supplementary data (Figure S7) and discussed in the manuscript on p. 14.

Remark: Even though there are some references in the introduction, results of table 2 should be discussed in regard with data from bibliography for relevant materials in the results and discussion section.

Answer: Additional discussion with a comparison of our results with already published ones was done on p.17.

Remark: Fig 9, again very low-quality figure and try to use totally different colors for depicting curves.

Answer: The quality of the figure was improved.

Reviewer 4 Report

This manuscript reports CO2 adsorption on the N- and P- ligand functionalized mesoporous silicas such as SBA-15 and MCM-48. The prepared pristine and functionalized mesoporous silica materials are characterized satisfactorily. The N and P functionalized silicas showed high adsorption capacity compared to the pristine materials. However, there are some issues that must be considered before accepting this paper for publication in Nanomaterials.

  1. Figure 4. The material may be MCM-48; errors must be corrected.
  2. The authors are advised to highlight the FTIR peaks in figure 4 and explain the potential bonding vibrational frequencies.
  3. At nearly 600 oC, the TGA of MCM-48/SAPTES revealed some unusual behavior that should be explained or corrected.
  4. Breakthrough figures for SBA-15 and MCM-18 can be simplified by splitting them into two separate figures. The analysis data points are merged.
  5. A comparison table of results must be included in the manuscript or supporting information.

Author Response

The authors thank the referees for the remarks and suggestions. The corrections needed have been made. The corrected version of the manuscript has been submitted.

Reviewer 4

This manuscript reports CO2 adsorption on the N- and P- ligand functionalised mesoporous silicas such as SBA-15 and MCM-48. The prepared pristine and functionalised mesoporous silica materials are characterized satisfactorily. The N and P functionalised silicas showed high adsorption capacity compared to the pristine materials. However, there are some issues that must be considered before accepting this paper for publication in Nanomaterials.

Remark: Figure 4. The material may be MCM-48; errors must be corrected.

Answer: The correction was made and we appologise for the mistake.

Remark: The authors are advised to highlight the FTIR peaks in figure 4 and explain the potential bonding vibrational frequencies.

Answer: The FTIR peaks in figure 4 (in the revised version Figure 3) were highlighted.

Remark: At nearly 600 oC, the TGA of MCM-48/SAPTES revealed some unusual behavior that should be explained or corrected.

Answer: The experiment was repeated and the new data have been inserted. The unusual behavior was attributed to the instrument electronic performance.

Remark: Breakthrough figures for SBA-15 and MCM-18 can be simplified by splitting them into two separate figures. The analysis data points are merged.

Answer: The figure was spitted in two. In the revised version it is Figure 9.

Remark: A comparison table of results must be included in the manuscript or supporting information.

Answer: A comparison table of our results with the already published ones was included in the manuscript – Table 3 on p. 18.

Round 2

Reviewer 2 Report

Now the paper is ready to be published in MDPI Nanomaterials

Reviewer 3 Report

Authors have made appropriate revisions.